# Predictors of Kidney Delayed Graft Function and Its Prognostic Impact following Combined Liver–Kidney Transplantation: A Recent Single-Center Experience

**DOI:** 10.3390/jcm11102724

**Published:** 2022-05-11

**Authors:** Paolo Vincenzi, Jeffrey J. Gaynor, Rodrigo Vianna, Gaetano Ciancio

**Affiliations:** 1Division of Transplant Surgery, Department of Surgery, Miami Transplant Institute, Jackson Memorial Hospital, University of Miami Miller School of Medicine, Miami, FL 33136, USA; paolo.vincenzi1981@gmail.com (P.V.); r.vianna@med.miami.edu (R.V.); 2Department of Surgery, Miami Transplant Institute, Jackson Memorial Hospital, University of Miami Miller School of Medicine, Miami, FL 33136, USA; jgaynor@med.miami.edu; 3Department of Urology, Jackson Memorial Hospital, University of Miami Miller School of Medicine, Miami, FL 33136, USA

**Keywords:** DGF, hypothermic pulsatile machine perfusion, combined liver and kidney transplantation, risk factors, graft survival, patient survival

## Abstract

Combined liver–kidney transplantation (CLKT) improves patient survival among liver transplant recipients with renal dysfunction. However, kidney delayed graft function (kDGF) still represents a common and challenging complication that can negatively impact clinical outcomes. This retrospective study analyzed the incidence, potential risk factors, and prognostic impact of kDGF development following CLKT in a recently transplanted cohort. Specifically, 115 consecutive CLKT recipients who were transplanted at our center between January 2015 and February 2021 were studied. All transplanted kidneys received hypothermic pulsatile machine perfusion (HPMP) prior to transplant. The primary outcome was kDGF development. Secondary outcomes included the combined incidence and severity of developing postoperative complications; development of postoperative infections; biopsy-proven acute rejection (BPAR); renal function at 1, 3, 6, and 12 months post-transplant; and death-censored graft and patient survival. kDGF was observed in 37.4% (43/115) of patients. Multivariable analysis of kDGF revealed the following independent predictors: preoperative dialysis (*p* = 0.0003), lower recipient BMI (*p* = 0.006), older donor age (*p* = 0.003), utilization of DCD donors (*p* = 0.007), and longer delay of kidney transplantation after liver transplantation (*p* = 0.0003). With a median follow-up of 36.7 months post-transplant, kDGF was associated with a significantly increased risk of developing more severe postoperative complication(s) (*p* < 0.000001), poorer renal function (particularly at 1 month post-transplant, *p* < 0.000001), and worse death-censored graft (*p* = 0.00004) and patient survival (*p* = 0.0002). kDGF may be responsible for remarkable negative effects on immediate and potentially longer-term clinical outcomes after CLKT. Understanding the important risk factors for kDGF development in CLKT may better guide recipient and donor selection(s) and improve clinical decisions in this increasing group of transplant recipients.

## 1. Introduction

Combined liver–kidney transplantation (CLKT) was first described as a treatment modality to overcome the development of end-stage liver disease (ESLD) concomitantly with end-stage kidney disease (ESKD) by transplanting both grafts from the same donor. CLKT has been increasingly performed since the Model for End-stage Liver Disease (MELD) liver allocation system was introduced in 2002 [1], prioritizing liver transplant candidates with associated renal dysfunction. In 2017, the United Network for Organ Sharing (UNOS) established the currently used medical eligibility criteria for CLKT in the United States [2]. CLKT now comprises approximately 10% of all liver transplant (LT) procedures in the United States [2]. 

For ESLD patients on dialysis at the time of transplant, graft and patient survival has been higher among CLKT recipients compared with patients undergoing liver transplant alone (LTA) or kidney after liver transplant (KALT), i.e., among those receiving a KT from a different donor at some point following the LTA [3,4]. Indeed, reduced renal allograft rejection following CLKT has been reported secondary to the immuno-protection granted to the kidney allograft by the transplanted liver of the same donor [5,6].

Nevertheless, kidney delayed graft function (kDGF) is a common and challenging postoperative complication in CLKT, with a reported incidence ranging from 7–50% in single-center studies [7,8,9] and larger UNOS data-derived analyses [10]. As in kidney-alone transplant recipients [11], kDGF has been associated with worse graft and patient survival following CLKT [2,8,9,10,12]. Several risk factors for kDGF development following CLKT have been reported, including various recipient characteristics, donor parameters, and methods of renal allograft preservation [2,7,8,9,12,13,14], all consistent with various reports in isolated KT [15,16,17,18].

At our center, hypothermic pulsatile machine perfusion (HPMP) has been used prior to transplant in all CLKT recipients. The few existing reports comparing HPMP vs. static cold storage (CS) of the donor kidney in CLKT have documented a significantly lower kDGF incidence when HPMP is applied [8,13], a result already well documented in KT [17,19,20].

Delaying implantation of the renal allograft in CLKT is also an approach that we take at our center. As previously described by others, stabilization of the patient’s coagulopathy and hemodynamics by weaning vasopressor support are benefits of the delayed strategy compared with performing a simultaneous liver–kidney (SLK) transplant [7,13]. The delayed approach therefore allows the surgeon to optimize the in vivo environment for the new kidney and to minimize blood loss during KT, reducing post-transplant complications in spite of a significantly prolonged cold ischemia time (CIT) for the renal allograft [7,13].

The purpose of this observational study was therefore to determine which subset of baseline variables were most influential in predicting kDGF development as well as to determine the prognostic impact of kDGF development in our unique CLKT cohort (i.e., simultaneous use of HPMP combined with delayed implantation of the renal allograft when deemed necessary).

## 2. Materials and Methods

### 2.1. Study Design and Endpoints

A retrospective cohort study was conducted on 115 consecutive adult patients undergoing CLKT at the Miami Transplant Institute between 1 January 2015 and 28 February 2021. The decision to include patients transplanted since 2015 was based on adoption of uniform and homogenous criteria for CLKT listing at our Institute, also in agreement with those introduced by UNOS in 2017 [2]. These criteria apply to both cirrhotic patients with chronic kidney disease (CKD) and with acute kidney injury (AKI) that, in the setting of chronic liver failure, is mainly sustained by type 1 hepatorenal syndrome (HRS). 

Data were obtained from a prospectively maintained electronic database and complemented by a review of clinical charts and donor data. The study was in accordance with the University of Miami Institutional Review Board and Helsinki Declaration. Written informed consent was obtained from all subjects or a legal surrogate. Last follow-up date for this study was 31 July 2021.

The primary study endpoint was the incidence of kDGF, defined as the requirement for dialysis during the first week post-kidney transplant [11]. Secondary study endpoints included: doses of intravenous antithymocyte globulin that were given; incidence of kidney primary nonfunction (kPNF), defined as permanent loss of kidney graft function starting from the time of transplant with ongoing hemodialysis requirement at 3 months post-transplant [9]; renal allograft futility (RAF), defined as patient death or ongoing hemodialysis requirement at 3 months post-transplant [9]; renal allograft function at 1, 3, 6, and 12 months post-transplant using serum creatinine (sCr) and estimated glomerular filtration rate (eGFR) as defined by the Modification of Diet in Renal Disease (MDRD) formula [21]; the occurrence of biopsy-proven acute rejection (BPAR) according to the Banff criteria [22,23]; the occurrence of any postoperative hospital-acquired infections that required treatment [24]; the occurrence of postoperative complications and their severity as measured by Clavien–Dindo grade and the Comprehensive Complication Index (CCI) [25,26]; length of hospital stay; death-censored graft failure (defined as the return to permanent dialysis or graft nephrectomy) [27]; and patient survival. 

In accordance with the Clavien–Dindo classification [25], a single organ dysfunction such as requirement for temporary dialysis is codified as a grade IVa complication. Since attributing a grade IVa classification to patients with kDGF solely because of their requirement for temporary dialysis during the first postoperative week would likely increase the association of kDGF incidence with postoperative morbidity risk, we decided not to include kDGF development (by itself) in the assessment of postoperative morbidity. We believe that this approach provided a more unbiased assessment of the association between kDGF incidence and postoperative morbidity risk. 

International Club of Ascites (ICA) criteria were used for the diagnosis and staging of acute kidney injury (AKI), regardless of the cause [28]. 

### 2.2. Hypothermic Pulsatile Machine Perfusion

Following kidney recovery, all deceased donor kidneys were connected to the LifePort Renal Preservation Machine^®^ (Organ Recovery Systems, Itasca, IL, USA), a type of non-oxygenated hypothermic machine perfusion. Renal allografts were arterially cannulated and perfused with Kidney Perfusion Solution^®^ (KPS-1). KPS-1, having the same composition as UW^®^ Machine Perfusion Solution, originally formulated at the University of Wisconsin, is a clear, sterile, non-pyrogenic, non-toxic solution with an osmolality of 300 ± 15 mOsm/kg, a sodium concentration of 100 mEq/L, a potassium concentration of 25 mEq/L, and a pH of 7.4 ± 0.1 at room temperature. Based on the sodium/potassium ratio, the composition is thus consistent with that of an extracellular solution [29].

Perfusion temperature was set at 4 °C, and systolic perfusion pressure was initially fixed at 30 mm Hg, with adjustment made after 30 min to obtain a flow goal of >100 mL/min. Pumping flow (mL/min) and resistive indices (RIs) (mmHg/mL/min) were serially monitored. Kidneys were maintained under these conditions at a centralized holding facility at the Miami Transplant Institute until the time of transplant, in order to facilitate the logistics related to CLKT, independent of whether the delayed or simultaneous strategy was used [30,31].

Hypothermic perfusion was not applied to liver allografts.

### 2.3. Transplant Protocol

All LT recipients at our center undergo a second post-transplant operation between postoperative days 2–3 (only the skin is closed at the time of LT). During the second-look operation, the liver allograft is evaluated, a biopsy is generally obtained, and an accurate washout of the abdomen is performed to evacuate any existing fluid collection or hematoma along with final closure of the abdominal wall. This method aims at minimizing the risk of developing graft congestion/abdominal compartment syndrome in the immediate post-reperfusion period, and in assessing the early occurrence of bile duct necrosis (or leak) and/or other unrecognized bowel injuries or obstructions [32].

By performing the standard second-look operation after LT as described above, it was not automatically included among the postoperative complications, as this approach represents a surgical protocol at our Institute. However, if any other procedures in addition to those listed above were performed, then the second look operation was classified as a grade IIIb postoperative complication, according to the Clavien–Dindo scale.

All LT cases in our series were performed using the piggyback hepatectomy technique without employing a veno-venous bypass.

KT was always performed using the conventional extra-peritoneal approach. The right iliac fossa was the site for all primary kidney allograft implantations; re-transplant operations were completed contralaterally. After completing the vascular anastomoses, a modified extravesical ureteroneocystostomy technique was used in all recipients [33].

Primary fascia closure was always accomplished at the end of KT. 

In all cases of delayed KT, continuous veno-venous hemodialysis (CVVHD) was used during the time transpiring between performance of the LT and KT.

All recipients received immunosuppressive therapy according to our center protocols, with induction consisting of at least one dose of intravenous antithymocyte globulin (ATG) (1 mg/kg) and basiliximab (20 mg) plus 3 or 4 doses of methylprednisolone (500 mg) [34,35]. The first induction dose of methylprednisolone alone was given intraoperatively before reperfusion of the liver allograft, whereas the first induction doses of ATG and basiliximab were administered intraoperatively before reperfusion of the renal allograft. Additional doses of ATG and basiliximab for a maximum total amount of 3 and 2 doses, respectively, were administered according to recipient clinical conditions, HLA sensitization, age, and race [34,35]. 

The dual induction therapy with ATG and the IL2rAb basiliximab, compared with ATG alone, results in a prolonged depletion, extended to 3 months post-transplant of CD25 cells that represent the alloantigen-stimulated effector T cells [34,35,36,37,38,39]. At the same time, the low dose of ATG applied in our protocol (maximum total dose of 3 mg/kg) is associated with a lower rate of rejection and viral infection than the standard regimen of ATG induction that adopts a total dose of 5 to 6 mg/kg, being of particular benefit to combined liver and kidney recipients at high risk of delayed graft function, which may not tolerate higher doses of ATG because of concerns about over-immunosuppression and concomitant risk of infections [34,35,36,37,38,39].

Achieving early and effective lymphocyte depletion with dual induction therapy may allow for the delayed introduction of calcineurin inhibitors, early withdrawal or complete avoidance of maintenance steroids, possible lower daily dosages of maintenance immunosuppressive agents, and reduced costs [34,35,36,37,38,39].

For highly sensitized recipients, i.e., those with a calculated panel reactive antibody ≥80%, low level positive flow crossmatch and/or high levels of preformed donor specific antibodies (MFI ≥ 6000), Rituximab (375 mg/m^2^ × 1 dose) was also given. 

Maintenance immunosuppression consisting of tacrolimus and mycophenolate mofetil started on postoperative day 1. Maintenance steroids were only used in highly sensitized recipients, according to our protocol [34]. 

### 2.4. Statistical Analysis

Frequency distributions were determined for baseline categorical variables, and the arithmetic mean along with standard error (±SE) were calculated for baseline continuous variables (with geometric mean and corresponding SE being used for baseline continuous variables having skewed distributions). 

Tests of association between baseline variables and kDGF development (No: kIGF (immediate graft function)/Yes: kDGF) were performed using Pearson (uncorrected) chi-squared tests for dichotomous baseline variables and standard *t*-tests for continuous baseline variables (using natural logarithmic transformed values for skewed distributions). Tests of association between kDGF development and various outcome variables were performed using Pearson (uncorrected) chi-squared tests for dichotomous outcomes, standard *t*-tests for continuous outcome variables, and log-rank tests for time-to-event outcomes. Time-to-event graphs were performed using the Kaplan–Meier technique.

Of note, there were some patients that had eGFR values much larger than 120 mL/min/1.73 m^2^ at various times post-transplant; thus, for the purpose of achieving greater statistical precision and power, all eGFR values ≥120 were capped at 120 mL/min/1.73 m^2^ in the statistical analyses. 

Multivariable analysis was performed using stepwise logistic and linear regression. In the attempt to avoid the selection of potentially spurious associations into these multivariable models, a more stringent type I error of 0.025 was used. All of the statistical analysis was performed using the Statistical Analysis System (SAS) software, Version 9.4, SAS Institute, Inc., Cary, NC, USA.

## 3. Results

### 3.1. Distributions of Baseline Characteristics

Overall, the mean (±SE) recipient age was 59.6 (±0.9) years; the percentage of male recipients was 60.0% (69/115). Recipient race/ethnicity included: 52.2% (60/115) being Hispanic, 31.3% (36/115) being white, 13.9% (16/115) being African-American, and 2.6% (3/115) being Asian.

Non-alcoholic steato-hepatitis (*n* = 39, 33.9%), chronic hepatitis C (*n* = 25, 21.7%), and alcoholic hepatitis (*n* = 23, 20.0%) represented the most frequent indications for LT, followed by cryptogenic cirrhosis including autoimmune hepatitis, primary sclerosing cholangitis, and primary biliary cirrhosis (*n* = 20, 17.3%); autosomal dominant polycystic liver disease (*n* = 9, 7.8%); and chronic hepatitis B (*n* = 6, 5.2%). Hepatocellular carcinoma was diagnosed in the explant pathology in 15 patients (13.0%). A combination of at least two different etiologic agents of ESLD was reported in seven patients (6.0%). 

The most common indication for KT was diabetic and/or hypertensive nephropathy (*n* = 62, 53.9%) followed by HRS (*n* = 47, 40.9%), autosomal dominant polycystic kidney disease (*n* = 14, 12.2%), and chronic glomerulonephritis (*n* = 8, 7%). HCV-associated glomerulonephritis, HIV-associated nephropathy, calcineurin-inhibitor induced nephropathy, and chronic rejection were the other etiologies of CKD in 20.9% of recipients (*n* = 24). A combination of at least two different etiologic agents of ESKD was reported in 40 patients (41.1%). 

Of note, six (5.2%) and five (4.3%) CLKT recipients already underwent previous isolated liver and isolated kidney transplants, respectively. 

Mean BMI at listing was significantly higher in recipients with NASH cirrhosis (30.0 ± 0.8) compared to recipients with alcoholic cirrhosis (24.5 ± 0.9) and other causes of ESLD (27.4 ± 0.6) (*p* = 0.0002).

Mean allocation MELD score at listing was 28.7 (±0.6), while the physiologic MELD score at the time of CLKT was 28.5 (±0.7), with a median time on waitlist of 65 (range: 2–2405) days. Sustained isolated AKI, AKI on CKD, and isolated CKD were the criteria justifying the KT in 12 (10.4%), 34 (29.6%), and 69 (60.0%) recipients, respectively. 

Delayed implantation of the renal allograft was performed in 69.6% (80/115) of cases, with a mean delayed time among these 80 cases of 20.7 (±0.9) hours after the LT. Vasopressor support was required during the operation in 71 (61.7%) patients, while in 64 (55.7%) patients, it was used within the first 6 h after the LT. CVVHD was administered preoperatively in 22 (19.1%) cases with a geometric mean duration of 6.8 (*/1.3) days, intraoperatively in 78 (67.8%) cases, and postoperatively (after the KT) in 30 (26.1%) cases. 

Mean liver and kidney CITs were 5.3 (±0.1) and 23.8 (±1.1) hours, respectively. Mean donor age was 38.4 (±1.4) years, and DCD donors were used in 19 cases (16.5%). Mean Kidney Donor Profile Index (KDPI) [40] was 40.9 (±2.5) %. Lastly, means of 13.7 (±1.5) and 2.8 (±0.2) units of packed red blood cells (PRBC) were transfused during the LT and KT, respectively.

Regarding HPMP, the mean time lapse to initiation of HPMP was 3.93 (±0.22) hours, whereas the mean duration of pumping was 19.84 (±1.07) hours. The mean initial flow and RIs were 55.2 (±4.1) mL/min and 0.7 (±0.04) mmHg/mL/min, respectively, while mean final flow and RIs were 152.1 (±2.4) mL/min and 0.21 (±0.01) mmHg/mL/min, respectively. 

### 3.2. Distributions of Outcome Variables

In total, 43 patients (37.4%) developed kDGF. Eight patients (7.0%) had kPNF, while RAF was observed in 10.4% of recipients (*n* = 12). Most of the patients received 3 doses of antithymocyte globulin and 2 doses of basiliximab as induction treatment (80.9% and 88.7%, respectively). Postoperative hospital-acquired infections were reported in 56 (48.7%) recipients. In total, 67 recipients (58.3%) experienced a postoperative complication of Clavien–Dindo grade ≥3, while mean CCI was 51.4 (±2.5). Geometric mean hospital length of stay was 24.0 (*/1.1) days. 

A first BPAR episode was observed in 7 (6.1%) and 13 (11.3%) cases of the renal and hepatic allografts, respectively. All first episodes of kidney and liver acute rejection were T-cell mediated, grade IA and mild, respectively.

During a median follow-up of 36.7 (range: 3.8–74.7) months post-transplant, death-censored hepatic allograft failure was documented in 5 patients (4.3%), with a median time-to-occurrence of 2.9 (range: 1.8–45.1) months post-transplant. Death-censored renal allograft failure occurred in 14 patients (12.2%) with a median time-to-occurrence of 3.0 (range: 0.03–51.6) months post-transplant. Death was reported in 21 patients (18.3%) with a median time-to-occurrence of 8.1 (range: 0.7–58.6) months post-transplant, of whom 9 (42.9%) died with functioning allografts. 

The leading cause for recipient mortality after CLKT was sepsis with multiple organ system failure (*n* = 13, 61.9%), followed by cardiopulmonary failure (*n* = 4, 19.0%), liver allograft failure (*n* = 2, 9.5%), hemorrhagic stroke (*n* = 1, 4.8%), and an unknown reason (*n* = 1, 4.8%).

### 3.3. Univariable Comparisons of Baseline Variables between kIGF and kDGF

All preoperative recipient, preoperative donor/HPMP, and operative variables that were in included in the statistical analysis are listed in Table 1, Table 2 and Table 3, respectively.

Tests of association of recipient baseline variables with kDGF development found that the mean recipient BMI was significantly lower in the kDGF group (*p* = 0.002), a result that was consistent across the etiology of ESLD (specific results not shown), while patients on hemodialysis before transplant (and with longer hemodialysis duration) were more likely to develop kDGF (*p* = 0.0005 and 0.02, respectively) (Table 1). 

Regarding baseline donor variables, only the percentage of DCD grafts and mean Donor Risk Index (DRI) [41] differed between the two groups, with significantly more organs from donors after circulatory death and with higher mean DRI in the kDGF group (*p* = 0.002 and 0.01, respectively) (Table 2). 

Among HPMP parameters analyzed, mean final RIs were significantly higher in the kDGF group (*p* = 0.02) (Table 2). Mean time from LT to KT and, consequently, mean duration of perfusion and mean kCIT, were significantly longer among kDGF patients compared to those with kIGF (*p* = 0.0002 each), as displayed in the HPMP and operative variable comparisons (Table 2 and Table 3). Likewise, the delayed technique of kidney implantation was performed in 81.4% (35/43) of patients developing kDGF compared with 62.5% (45/72) of those with kIGF, approaching statistical significance (*p* = 0.03). Among the other operative variables analyzed, a significantly higher mean number of PRBC units were transfused during LT in patients with kDGF (*p* = 0.006).

### 3.4. Univariable Comparisons of Outcome Variables between kIGF and kDGF

In comparing postoperative outcomes by group (Table 4), recipients with kDGF generally had more impaired renal function post-transplant compared to kIGF patients, as evidenced by significantly worse mean eGFR and geometric mean sCr at both 1 month (*p* < 0.000001 each) and 3 months (*p* = 0.004 and 0.0004) post-transplant, respectively. Significant differences in geometric mean sCr at 6 and 12 months post-transplant showing poorer renal function in the kDGF group were observed, while similar but nonsignificant trends existed for mean eGFR (Table 4). However, when analyzing the percentage of patients who developed CKD stage ≥3B (i.e., eGFR < 45 mL/min/1.73 m^2^), significantly higher percentages were observed in the kDGF group at each time point (*p* < 0.000001, 0.0001, 0.008, and 0.02 at 1, 3, 6, and 12 months post-transplant, respectively).

Similarly, percentages developing RAF and kPNF were significantly higher in the kDGF group (*p* = 0.000002 and 0.0001, respectively). Freedom-from-death-censored renal allograft failure and patient survival were also significantly less favorable for patients who developed kDGF (*p* = 0.00004 and 0.0002, respectively), as shown in Table 4 and Figure 1 and Figure 2.

Regarding overall postoperative morbidity, the percentages of patients developing a major complication as classified by Clavien–Dindo grade ≥3 and any hospital-acquired infection were significantly higher in the kDGF group (*p* = 0.000003 and 0.002, respectively) (Table 4). Consequently, mean CCI was significantly higher (*p* < 0.000001) in the kDGF group, and length of hospital stay was significantly longer (*p* < 0.000001). 

No significant difference was observed in the percentage developing a BPAR of the renal allograft between the two groups (*p* = 0.27), even as recipients with kDGF received significantly fewer doses of anti-thymocyte globulin as induction treatment (*p* = 0.002), and the third dose (*p* = 0.0009) in particular (Table 4).

### 3.5. Multivariable Analysis Results

In a stepwise logistic regression analysis of baseline predictors of kDGF, five variables were selected containing independent predictive value: lower recipient BMI (*p* = 0.006), receiving preoperative dialysis (*p* = 0.0003), a prolonged time from LT to KT (*p* = 0.0003), use of a DCD graft (*p* = 0.007), and older donor age (*p* = 0.003) (Table 5). Note that the association of older donor age with a greater risk of developing kDGF became significant once the prognostic effect of time from LT to KT was controlled. Once these five multivariable predictors of kDGF were controlled, none of the other baseline variables offered additional predictive value.

Of note, three of the baseline variables were highly correlated/confounded: longer Time from Liver to Kidney Tx (hr), longer CIT Kidney, and longer HPMP Pump Time (*p* < 0.000001 for each pairwise test of linear correlation). Thus, it was statistically impossible to determine which of these three baseline variables was the most appropriate multivariable predictor of kDGF development (in Table 5). Given that {CIT Kidney—Time from Liver to Kidney Tx} and Static CS Time had no univariable associations with kDGF risk (*p* = 0.92 and 0.96, respectively), it appeared that, biologically, Time from Liver to Kidney Tx (hr) was the most appropriate variable to be selected into the kDGF logistic model, as Time from Liver to Kidney Tx (hr) clearly reflects the patient’s clinical status at the time of LT.

Multivariable analysis of the time from LT to KT revealed four independent baseline predictors of a longer time: younger donor age (*p* = 0.007), delayed implantation of the renal allograft (*p* < 0.000001), a greater number of units of PBRCs transfused during LT (*p* = 0.007), and postoperative vasopressor support within the first 6 h post-LT (*p* = 0.01) (Table 5).

Lastly, kDGF was a highly significant multivariable predictor of a higher CCI, even after controlling for the four other significant multivariable predictors of this outcome (*p* < 0.000001, Table 5). While kDGF was not an important multivariable predictor of the development of hospital-acquired infections (*p* = 0.05, Table 5), kDGF was a highly significant multivariable predictor of a poorer renal function at 1-month post-transplant (i.e., eGFR < 45 mL/min/1.73 m^2^), even after controlling for the 2 other significant multivariable predictors of this outcome (*p* < 0.000001, Table 5). Thus, the important univariable and multivariable associations of kDGF with less favorable clinical outcomes (as shown in Table 4 and Table 5, respectively) were similar.

## 4. Discussion

Our study represents one of the largest single-center series investigating baseline predictors of kDGF among CLKT recipients transplanted more recently (since 2015) and with routine use of HPMP for all donor kidneys. Among all the baseline variables assessed, we were able to identify specific risk factors for developing kDGF in CLKT recipients (lower recipient BMI, pretransplant recipient dialysis, use of DCD and older donors, and a longer time from liver to kidney transplant). The use of kidneys from donors after circulatory death and older donors was significantly associated with kDGF in the multivariable analysis, confirming several previous reports [10,12,14]. The prognostic significance of the other factors in predicting higher kDGF risk essentially agrees with previously published reports in both isolated KT [15,16,18] and in CLKT [2,7,8,9,10,13].

While the time elapsed between performing the LT and KT was observed to be a highly significant predictor of kDGF in our study, this finding should not lead one to conclude that delaying the kidney transplant might actually increase kDGF risk. In fact, a major consumption of blood products during the LT and the use of vasopressor support immediately after the LT were significant predictors of a longer time between the two operations in multivariable analysis, suggesting that higher degrees of surgical complexity during LT and recipient hemodynamic instability immediately following LT strongly support the approach of delaying the kidney transplant, at least for the sicker patients, an approach which has been promoted in other studies [7,13]. Thus, the prognostic value of longer time from LT to KT in terms of implying greater kDGF risk appears to be more a reflection of the patient’s clinical status at the time of LT rather than the longer waiting time for the KT itself. Similarly, kDGF might not by itself be a risk factor for the development of postoperative complications but actually could be a marker of hepatic ischemia/reperfusion injury, resulting in a more complicated postoperative course.

Similar conclusions regarding risk factors for kDGF were reached by Korayem et al. [8], stressing that the severity of illness of CLKT recipients at the time of LT is the main factor predisposing for increased kDGF incidence, worse short-term kidney allograft function, and inferior long-term graft survival. Nevertheless, in their study, only 12% of kidneys were placed on HPMP, in contrast to our using HPMP for all deceased donor kidneys, regardless of whether they were implanted simultaneously with the liver or later as a second operation (performed up to 48 h post-LT), with the latter option being our preferred approach, applied in roughly in 70% of cases, as in the series of Ekser et al. [7].

With the exceptions of our study and Ekser et al. [7], we are aware of no other reports in which HPMP was routinely used in patients receiving CLKT. According to Ekser et al. [7], the strategy of consistently applying machine perfusion and delaying transplantation of the renal allograft contributed to a significant reduction in kDGF and its associated morbidity risks, and it also helped to improve longer-term outcomes such as graft and patient survival, though future clinical studies should be addressed toward a tailored use of HPMP according to the donor parameters and the clinical conditions of the recipient at the time of transplant.

Recent evidence from a large randomized controlled trial [42] together with previous single-center studies [43,44,45] in KT indicate that longer CIT remains as an independent risk factor for DGF occurrence even when kidneys are preserved by HPMP. With each additional hour of cold ischemia reported to increase the odds of developing DGF by 8% [43], it would still be important, with all other considerations being equal, to limit CIT by as much as possible [42]. However, our study and Ekser et al. [7] confirm that, in the setting of CLKT, delayed implantation of the renal allograft when clinically indicated appears to represent a better strategy, even at the cost of prolonging kCIT.

Regarding the HPMP parameters recorded in our series, though the final resistive index was significantly higher in univariable analysis of kDGF risk, this association was not confirmed in multivariable analysis. Thus, its clinical significance might be limited here, as the final pumping indices reported were excellent in both groups (Table 2). Other indicators of donor quality such as the use of Extended Criteria Donor (ECD) [46] grafts, DRI, and KDPI did not show any significant associations in multivariable analysis of kDGF risk. This might be related to a low percentage of ECD grafts being used in our study population (13.0%) together with favorable mean DRI and KDPI being reported here (1.44 and 40.9%, respectively), demonstrating that acceptable quality grafts were allocated to CLKT recipients due to their relatively higher MELD scores. Indeed, regarding KDPI, the literature indicates a cut-off of 70% for increased risk of developing kDGF and worsening outcomes [14,47]. Similarly, the survival advantage conferred by CLKT compared to LTA decreases with the utilization of grafts with a higher KDPI, as demonstrated by Sharma et al. [48]. 

Lower recipient BMI was associated with a significantly higher risk of developing kDGF in our study. While the causes of ESLD were not directly associated with kDGF risk, patients with alcoholic cirrhosis had a significantly lower BMI. Thus, the association found here between lower recipient BMI and higher kDGF risk could be related to the clinical status of the patient at the time of LT. Clearly, it would be preferable to have included a measure of sarcopenia [49] before drawing any firm conclusions between kDGF and low BMI.

In terms of secondary endpoints, patients with kDGF had significantly worse renal function at 1, 3, 6, and 12 months post-transplant, in agreement with previous series conducted in isolated KT [50,51] and CLKT [8,9,13,48], as well as by our center [52]. Our study also showed a low incidence of kidney graft BPAR (6.1%) compared with the incidence reported in isolated KT [16,50,51]. This may be attributed to a protective effect of the liver allograft existing against cellular and antibody-mediated kidney rejection as shown by previous studies [5,6]. Concurrently, kDGF did not influence the occurrence of renal allograft rejection, though significantly fewer doses of the lymphocyte-depleting induction agent were given to recipients developing kDGF. A more vulnerable clinical status of these patients in the postoperative period might represent the underlying reason for their having received fewer induction doses. Lastly, significantly higher postoperative morbidity (higher Clavien–Dindo grade and CCI) was found among patients experiencing kDGF in our study.

There are several limitations to this study, particularly that of cause and effect. As previously described, patients with a difficult LT procedure might first have to be stabilized before they can receive the KT; thus, a longer time between LT and KT by itself might not be a risk factor, but a result of a complicated situation. Secondly, although the data were prospectively collected and stored in our hospital database, the analysis performed was a single-center retrospective cohort study. However, the recent time period in which all patients were transplanted (since 2015) and the use of well-established criteria for CLKT listing markedly reduced the likelihood that an “era effect” and/or selection bias would exist. Additionally, the same primary surgical team performed these CLKT transplants with few (if any) differences in transplant protocols, as the study period was relatively short. Single-center studies also allow for the collection and investigation of a larger number of potentially relevant baseline variables, which may contribute to a more accurate assessment of the associations between those variables and prespecified study outcomes.

Currently, at the Miami Transplant Institute, we do not have a standard protocol in place that determines the optimal timing of KT after LT during CLKT. By performing this retrospective study, our hope was to identify potential variables (such as the requirement for PBRCs during LT and post-LT requirement of vasopressor support) that could subsequently be used in establishing a more uniform approach to this growing group of solid organ transplants recipients.

## 5. Conclusions

The incidence of kDGF remains high among CLKT recipients that are known to have a high MELD score. Longer time from LT to KT resulted the strongest multivariable predictor of higher kDGF occurrence, followed by pre-transplant dialysis, older donor age, lower recipient BMI, and use of DCD grafts. Additionally, kDGF was associated with inferior post-transplant outcomes and lower graft and patient survival in this class of solid organ transplant recipients. Further prospective studies with larger number of patients undergoing CLKT are required to validate and confirm these findings.

## Figures and Tables

**Figure 1 jcm-11-02724-f001:**
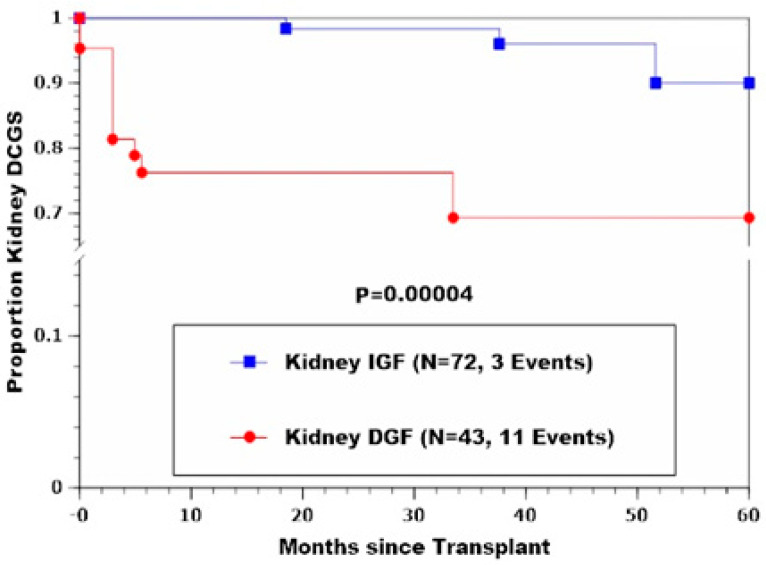
Kidney death-censored graft survival (DGCS) in kIGF and kDGF recipients showing significantly higher rate of graft loss among kDGF patients.

**Figure 2 jcm-11-02724-f002:**
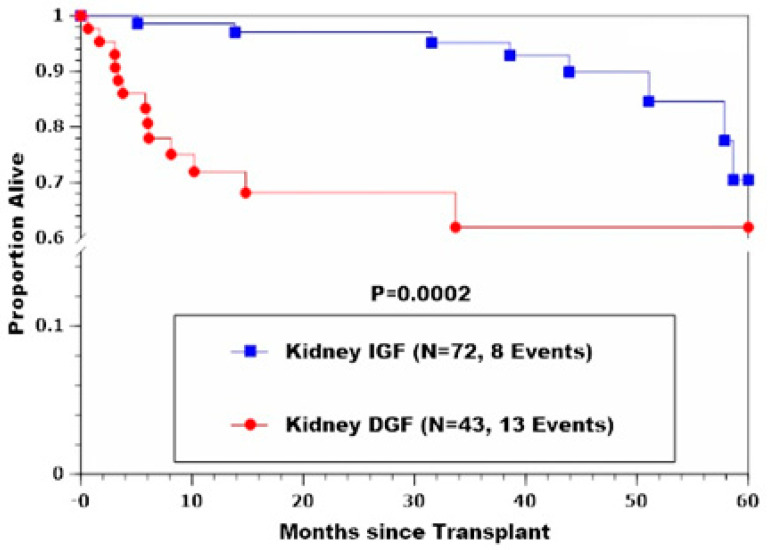
Patient survival in kIGF and kDGF recipients showing inferior survival rates among kDGF patients, particularly during the first few years post-CLKT.

**Table 1 jcm-11-02724-t001:** Comparisons of preoperative recipient variables by group (kIGF vs. kDGF) ^1^.

	kIGF Group (*n* = 72)	kDGF Group (*n* = 43)	*p* Value
**Recipient data**			
Age (years), mean ± SE	60.4 ± 1.1	58.4 ± 1.6	0.31
Male, % (*n*)	54.2 (39)	69.8 (30)	0.10
BMI (kg/m^2^), mean ± SE	28.9 ± 0.6	25.8 ± 0.6	**0.002**
*Race*, % (*n*)			0.48
Afro-American	12.5 (9)	16.3 (7)	
Hispanic	29.2 (21)	34.9 (15)	
Caucasian	56.9 (41)	44.2 (19)	
Other	1.4 (1)	4.7 (2)	
MELD score at listing, mean ± SE	28.8 ± 0.8	28.6 ± 1.1	0.83
MELD score at transplant, mean ± SE	28.4 ± 0.9	28.7 ± 1.1	0.80
Waitlist time(days), geometric mean */SE	65.1 */1.2	61.6 */1.2	0.84
DM, % (*n*)	58.3 (42)	69.8 (30)	0.22
HTN, % (*n*)	61.1 (44)	60.5 (26)	0.95
*Liver cirrhosis etiology*, % (*n*)			
Alcoholic	18.1 (13)	23.3 (10)	0.50
NASH	33.3 (24)	34.9 (15)	0.87
Hepatitis C	26.4 (19)	14.0 (6)	0.12
Cryptogenic/Autoimmune	18.1 (13)	16.2 (7)	0.81
HCC	15.3 (11)	9.3 (4)	0.36
Other °	12.5 (9)	14.0 (6)	0.82
Combination of ≥ 2 etiologic agents	6.9 (5)	4.6 (2)	0.61
*ESKD etiology*, % (*n*)			
HRS	41.7 (30)	39.5 (17)	0.82
Metabolic (DM and/or HTN)	52.8 (38)	55.8 (24)	0.75
ADPKD	12.5 (9)	11.6 (5)	0.89
Glomerulonephritis	4.2 (3)	11.6 (5)	0.13
Other °°	23.6 (17)	16.3 (7)	0.35
Combination of ≥ 2 etiologic agents	37.5 (27)	30.2 (13)	0.42
*Timing of Kidney Failure*, % (*n*)			
AKI	9.7 (7)	11.6 (5)	0.75
AKI on CKD	30.6 (22)	27.9 (12)	0.76
CKD	59.7 (43)	60.5 (26)	0.94
*Dialysis modality*			
Pre-emptive, % (*n*)	36.1 (26)	7.0 (3)	**0.0005**
Hemodialysis, % (*n*)	63.9 (46)	93.0 (40)	**0.0005**
HD duration (days), mean ± SE	268.4 ± 68.1	585.9 ± 124.6	**0.02**
Pre-transplant CVVHD, % (*n*)	13.9 (10)	27.9 (12)	0.06
Liver re-transplant, % (*n*)	5.6 (4)	4.7 (2)	0.83
Kidney re-transplant, % (*n*)	1.4 (1)	9.3 (4)	0.04
Preoperative hospitalization, % (*n*)	26.4 (19)	39.5 (17)	0.14
Hospitalization duration(days), mean ± SE	6.4 ± 1.9	9.0 ± 2.7	0.42
Preoperative ICU stay, % (*n*)	19.4 (14)	32.6 (14)	0.11
ICU stay duration (days), mean ± SE	2.4 ± 0.8	4.4 ± 1.7	0.22
RRI score, geometric mean */SE	7.4 */1.1	9.3 */1.1	0.08
cPRA ≥ 40%, % (*n*)	33.3 (24)	25.6 (11)	0.38
Preformed DSAs (class I or II), % (*n*)	16.7 (12)	20.9 (9)	0.57
Positive Crossmatch, % (*n*)	13.9 (10)	20.9 (9)	0.33

° Hepatitis B induced cirrhosis, Autosomic Dominant Polycystic Liver Disease. °° Calcineurin Inhibitor Toxicity, Focal Segmental Glomerulosclerosis, HCV or HIV nephropathy, Chronic Rejection kIGF, kidney Immediate Graft Function; kDGF, kidney Delayed Graft Function; BMI, Body Mass Index; MELD, Model for End-stage Liver Disease; DM, Diabetes Mellitus; HTN, Hypertension; NASH, Non-Alcoholic Steato-Hepatitis; HCC, Hepatocellular Carcinoma; ESKD, End-Stage Kidney Disease; HRS, Hepato-Renal Syndrome; ADPKD, Autosomic Dominant Polycystic Kidney Disease; AKI, Acute Kidney Injury; CKD, Chronic Kidney Disease; HD, Hemodialysis; CVVHD, Continuous Venous-Venous Hemodialysis; ICU, Intensive Care Unit; RRI, Renal Risk Index; cPRA, calculated Panel Reactive Antibody; DSAs, Donor Specific Antibodies. ^1^ Mean ± SE if continuous (Geometric Mean */SE for skewed distributions); Percentage with characteristic if categorical.

**Table 2 jcm-11-02724-t002:** Comparisons of preoperative donor/HPMP variables by group (kIGF vs. kDGF) ^1^.

	kIGF Group (*n* = 72)	kDGF Group (*n* = 43)	*p*-Value
**Donor data**			
Age (years), mean ± SE	36.6 ± 1.8	41.4 ± 2.1	0.10
BMI (kg/m^2^), mean ± SE	25.3 ± 0.5	26.6 ± 0.9	0.17
*Race*, % (*n*)			0.41
Afro-American	16.7 (12)	18.6 (8)	
Hispanic	27.8 (20)	34.9 (15)	
Caucasian	55.6 (40)	44.2 (19)	
Other	0 (0)	2.3 (1)	
DCD, % (*n*)	8.3 (6)	30.2 (13)	**0.002**
KDPI (%), mean ± SE	37.7 ± 3.0	46.2 ± 4.2	0.10
ECD, % (*n*)	9.7 (7)	18.6 (8)	0.17
DRI, mean ± SE	1.37 ± 0.04	1.55 ± 0.06	**0.01**
DM, % (*n*)	2.8 (2)	0.0 (0)	0.27
HTN, % (*n*)	11.1 (8)	25.6 (11)	0.04
*Cause of death*, % (*n*)			
Anoxia	38.9 (28)	41.9 (18)	0.75
Stroke	18.1 (13)	23.3 (10)	0.50
Head Trauma	41.7 (30)	34.9 (15)	0.47
Other °	1.3 (1)	0.0 (0)	0.44
*Vasopressor support* (*n*), % (*n*)			0.03
0	6.9 (5)	25.6 (11)	
1	33.3 (24)	18.6 (8)	
2	45.8 (33)	37.2 (16)	
3	13.9 (10)	18.6 (8)	
Terminal UO (mL/min), mean ± SE	146.3 ± 16.3	136.6 ± 11.2	0.67
Terminal sCr (mg/dL), geometric mean */SE	0.77 */1.05	0.89 */1.09	0.12
*Pre-implant kidney biopsy*, % (*n*)	83.3 (60)	90.7 (39)	0.27
(1) Glomerulosclerosis, mean ± SE	5.4 ± 0.7	4.2 ± 0.6	0.23
(2) Chronic Tubulo-Interstitial Injury			0.46
None, % (*n*)	31.7 (19)	20.5 (8)	
Minimal, % (*n*)	16.7 (10)	15.4 (6)	
Mild, % (*n*)	50.0 (30)	64.1 (25)	
Moderate, % (*n*)	1.7 (1)	0.0 (0)	
(3) Arteriolosclerosis			0.13
None, % (*n*)	38.3 (23)	25.6 (10)	
Minimal, % (*n*)	5.0 (3)	17.9 (7)	
Mild, % (*n*)	55.0 (33)	56.4 (22)	
Moderate, % (*n*)	1.7 (1)	0.0 (0)	
(4) Acute Tubular Necrosis			0.41
None, % (*n*)	3.3 (2)	0.0 (0)	
Minimal, % (*n*)	1.7 (1)	2.6 (1)	
Mild, % (*n*)	86.7 (52)	94.9 (37)	
Moderate, % (*n*)	8.3 (5)	2.6 (1)	
**HPMP data**			
Timing of initiation (hr), mean ± SE	3.93 ± 0.31	3.95 ± 0.29	0.96
Duration of perfusion (hr), mean ± SE	16.83 ± 1.23	24.88 ± 1.77	**0.0002**
Initial Flows (mL/min), mean ± SE	52.3 ± 4.7	60.2 ± 7.5	0.35
Initial Resistance (mmHg/mL/min), mean ± SE	0.74 ± 0.05	0.62 ± 0.05	0.15
Final Flows (mL/min), mean ± SE	153.1 ± 2.9	150.3 ± 4.4	0.59
Final Resistance (mmHg/mL/min), mean ± SE	0.20 ± 0.01	0.23 ± 0.01	**0.02**

° Bacterial meningitis. HPMP, Hypothermic Pulsatile Machine Perfusion; kIGF, kidney Immediate Graft Function; kDGF, kidney Delayed Graft Function; BMI, Body Mass Index; DCD, Donation after Circulatory Death; KDPI, Kidney Donor Profile Index; ECD, Extended Criteria Donor; DRI, Donor Risk Index; DM, Diabetes Mellitus; HTN, Hypertension; UO, Urinary Output; sCr, serum Creatinine. ^1^ Mean ± SE if continuous (Geometric Mean */SE for skewed distributions); Percentage with characteristic if categorical.

**Table 3 jcm-11-02724-t003:** Comparisons of operative variables by group (kIGF vs. kDGF) ^1^.

	kIGF Group (*n* = 72)	kDGF Group (*n* = 43)	*p*-Value
Time from LT to KT (hr), mean ± SE	11.4 ± 1.2	19.5 ± 1.8	**0.0002**
Delayed KT, % (*n*)	62.5 (45)	81.4 (35)	0.03
CIT liver (hr), mean ± SE	5.4 ± 0.1	5.1 ± 0.2	0.38
WIT liver (min), mean ± SE	26.3 ± 0.7	27.7 ± 0.7	0.17
CIT kidney (hr), mean ± SE	20.7 ± 1.2	28.8 ± 1.8	**0.0002**
WIT kidney (min), mean ± SE	29.8 ± 1.0	29.9 ± 1.6	0.95
PRBC units during LT, mean ± SE	10.5 ± 1.1	19.2 ± 3.6	**0.006**
PRBC units during KT, mean ± SE	2.4 ± 0.2	3.3 ± 0.4	0.04
Vasopressor support, *n* (%)	59.7 (43)	65.1 (28)	0.56

kIGF, kidney Immediate Graft Function; kDGF, kidney Delayed Graft Function; LT, Liver Transplant; KT, Kidney Transplant; CIT, Cold Ischemic Time; WIT, Warm Ischemic Time; PRBC, Packed Red Blood Cell. ^1^ Mean ± SE if continuous; Percentage with characteristic if categorical.

**Table 4 jcm-11-02724-t004:** Comparisons of postoperative outcomes by group (kIGF vs. kDGF) ^1^.

	kIGF Group (*n* = 72)	kDGF Group (*n* = 43)	*p*-Value
Vasopressor support, % (*n*)	52.8 (38)	60.5 (26)	0.42
**sCr (mg/dL), geometric mean */SE** °			
at 1 mo post-transplant	0.88 */1.04 (*n* = 72)	1.72 */1.13 (*n* = 39)	**<0.000001**
at 3 mo post-transplant	0.95 */1.04 (*n* = 72)	1.35 */1.12 (*n* = 36)	**0.0004**
at 6 mo post-transplant	1.01 */1.03 (*n* = 70)	1.31 */1.08 (*n* = 32)	**0.0005**
at 12 mo post-transplant	1.05 */1.03 (*n* = 69)	1.35 */1.08 (*n* = 23)	**0.0006**
**MDRD-eGFR (mL/min/1.73 m^2^), mean ± SE** °			
at 1 mo post-transplant	84.2 ± 3.1	50.4 ± 6.0	**<0.000001**
at 3 mo post-transplant	79.2 ± 3.1	61.8 ± 5.6	**0.004**
at 6 mo post-transplant	73.3 ± 2.7	61.9 ± 5.2	0.03
at 12 mo post-transplant	69.1 ± 2.4	57.9 ± 4.6	0.03
**CKD stage ≥ 3B (eGFR < 45 mL/min/1.73 m^2^), % (*n*)** °			
at 1 mo post-transplant	6.9 (5/72)	66.7 (26/39)	**<0.000001**
at 3 mo post-transplant	6.9 (5/72)	36.1 (13/36)	**0.0001**
at 6 mo post-transplant	10.0 (7/70)	31.3 (10/32)	**0.008**
at 12 mo post-transplant	10.1 (7/69)	30.4 (7/23)	**0.02**
RAF, % (*n*)	0.0 (0)	27.9 (12)	**0.000002**
kPNF, % (*n*)	0.0 (0)	18.6 (8)	**0.0001**
Kidney BPAR, % (*n*)	4.2 (3)	9.3 (4)	0.27
**Induction treatment**			**0.002**
Anti-thymocyte globulin doses, % (*n*)			
1	4.2 (3)	25.6 (11)	
2	5.6 (4)	9.3 (4)	
3	90.3 (65)	65.1 (28)	**0.0009**
Death-censored renal allograft failure, % (*n*)	4.2 (3)	25.6 (11)	**0.00004 ^2^**
Death, % (*n*)	11.1 (8)	30.2 (13)	**0.0002 ^2^**
**Morbidity**			
Clavien-Dindo grade			**<0.000001**
I, % (*n*)	6.9 (5)	0.0 (0)	
II, % (*n*)	51.4 (37)	14.0 (6)	
IIIA, % (*n*)	6.9 (5)	7.0 (3)	
IIIB, % (*n*)	22.2 (16)	9.3 (4)	
IVA, % (*n*)	9.7 (7)	32.6 (14)	
IVB, % (*n*)	2.8 (2)	18.6 (8)	
V, % (*n*)	0.0 (0)	18.6 (8)	
Clavien-Dindo grade ≥ III, % (*n*)	41.7 (30)	86.0 (37)	**0.000003**
CCI, mean ± SE	39.2 ± 2.2	71.9 ± 3.7	**<0.000001**
Hospital-acquired infections, % (*n*)	37.5 (27)	67.4 (29)	**0.002**
Length of hospital stay (days), geometric mean */SE	16.5 */1.09	45.1 */1.14	**<0.000001**

° The serum Cr and eGFR for a patient who previously developed renal allograft failure were not imputed here; therefore, those patients were not utilized in these calculations. kIGF, kidney Immediate Graft Function; kDGF, kidney Delayed Graft Function; sCr, serum Creatinine; MDRD-eGFR, Modification of Diet in Renal Disease-estimated Glomerular Filtration Rate; CKD, Chronic Kidney Disease; RAF, Renal Allograft Futility; kPNF, kidney Primary Non-Function; BPAR, Biopsy-Proven Acute Rejection; CCI, Comprehensive Complication Index. ^1^ Mean ± SE if continuous (Geometric Mean */SE for skewed distributions); Percentage with characteristic if categorical. ^2^
*p*-value based on the log-rank test.

**Table 5 jcm-11-02724-t005:** Multivariable Analysis Results.

**(a)** **Preoperative Risk Factors Contributing to kDGF** **Variable**	***p*-Value**	**Logistic Regression** **Model Coefficient ± SE**
Recipient BMI	0.006	−0.142 ± 0.055
Pre-transplant HD	0.0003	2.489 ± 0.766
Time from LT to KT	0.0003	0.084 ± 0.025
DCD graft	0.007	1.829 ± 0.709
Donor age	0.003	0.052 ± 0.019
**(b)** **Predictors of Time from LT to KT** **Variable**	***p*-value**	**Linear Regression** **Model Coefficient ± SE**
Donor age	0.007	−0.108 ± 0.039
Delayed KT	<0.000001	18.876 ± 1.329
PRBC units during LT	0.007	0.102 ± 0.037
Postoperative vasopressor support	0.01	3.102 ± 1.218
**(c)** **Predictors of CCI** **Variable**	***p*-value**	**Linear Regression** **Model Coefficient ± SE**
Pre-transplant HD	0.002	14.914 ± 4.701
Time from LT to KT	0.004	0.580 ± 0.195
PRBC units during LT	0.02	0.314 ± 0.134
Postoperative vasopressor support	0.006	12.249 ± 4.370
Positive Association of kDGF with CCI °	<0.000001	
**(d)** **Predictors of Hospital-acquired infections** **Variable**	***p*-value**	**Logistic Regression** **Model Coefficient ± SE**
Pre-transplant ICU stay	0.004	1.373 ± 0.493
Time from LT to KT	0.003	0.052 ± 0.018
Positive Association of kDGF with hospital-acquired infections °°	0.05	
**(e)** **Predictors of CKD stage ≥ 3B (eGFR < 45 mL/min/1.73 m^2^) at 1 mo** **Variable**	***p*-value**	**Logistic Regression** **Model Coefficient ± SE**
DCD graft	0.01	1.319 ± 0.551
RPM final RIs	0.0002	13.268 ± 4.150
Positive Association of kDGF with CKD stage ≥ 3B at 1 mo °°°	<0.000001	

° after controlling for the 4 selected independent baseline predictors of this outcome variable. °° after controlling for the 2 selected independent baseline predictors of this outcome variable. °°° after controlling for the 2 selected independent baseline predictors of this outcome variable. kDGF, kidney Delayed Graft Function; BMI, Body Mass Index; HD, Hemodialysis; LT, Liver Transplant; KT, Kidney Transplant; DCD, Donation after Circulatory Death; PRBC, Packed Red Blood Cell; CCI, Comprehensive Complication Index; ICU, Intensive Care Unit; CKD, Chronic Kidney Disease; eGFR, estimated Glomerular Filtration Rate; RPM, Renal Perfusion Machine; RIs, Resistive Indices.

## Data Availability

All the data are deposited in a database stored at the Miami Transplant Institute.

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
