# Peer review of "Predictors of Kidney Delayed Graft Function and Its Prognostic Impact following Combined Liver–Kidney Transplantation: A Recent Single-Center Experience"

_jcm, 2022, doi:10.3390/jcm11102724_

Round 1

Reviewer 1 Report

Review JCM Vicenzi et al

Predictors of Kidney Delayed Graft Function and Its Prognostic 2 Impact following Combined Liver-Kidney Transplantation: A 3 Recent Single-Center Experience

In this paper the authors have investigated the predictors of kidney DGF and its impact in combined liver kidney transplantation using a single center analysis. Using the data of 115 patients where all kidneys were transplanted after hypothermic pulsatile machine perfusion (HPMP). Around one third developed kDGF and one of the risk factors was longer delay of KT after initial liver transplantation. kDGF was associated with more severe postoperative complications, worse long/term kidney function and worse graft and patient survival. This data including the use of HPMP in combined LKT is interesting, but I do have several concerns regarding the methods that were used and the conclusion that were drawn.

  • The main issue of this analysis is cause and effect. Patients with a difficult LT procedure might first have to be stabilized before they can receive the KT. As described by the authors, a longer time between LT and KT by itself might not be a risk factor, but a result of a complicated situation. Furthermore, kDGF might not be a risk factor for postoperative complications, but actually be a ‘ marker’ of hepatic ischemia/reperfusion injury, resulting in a complicated postoperative course.
  • Can the authors describe the protocol regarding timing of KT in combined LKT. Is there a standard period? And what are the reasons for delaying a KT?
  • The introduction is relatively long, i.e. a detailed description of HPMP can be given in the methods or discussion.
  • There are multiple writing errors in the text, correction for English language is required. ‘ Primary outcome’ is used for the comparison of interventions in clinical trials, not observational studies. The conclusion kDGF is ‘ responsible ’ for … is somewhat premature for the results presented in this retrospective study.
  • Please delete the table with the included variables in this study. This could be described in the text. The variables will be shown in the tables with the results already.
  • Regarding machine perfusion: All kidneys were perfused with HPMP. Was this oxygenated perfusion and what kind of solution was used, please describe KPS. Were the livers in these patients also perfused? Please describe. AS perfusion of the liver might decrease the hepatic ischemia/reperfusion injury, this might also affect injury to the kidneys in the early post transplant period.
  • Can the authors also describe the technicalities of the liver transplant procedure? classic, piggyback, venovenous bypass? This could all affect kidney function after LT.
  • The authors describe a standard second look in their liver transplant patients, is this also for liver only patients? If any intervention is performed during this procedure, does it count as a postoperative complication?
  • The authors describe the immunosuppression starting at KT. Did the patients not receive their first Immunosuppression when the liver was transplanted?
  • The table with 5 (!) multivariable analyses is very confusing and has limited value. The authors just summarized the results, but failed to describe the additional value of each analysis.

Author Response

Point-by-Point Responses to the Reviewers’ Comments

Reviewer #1’s Comments

1. “The main issue of this analysis is cause and effect. Patients with a difficult LT procedure might first have to be stabilized before they can receive the KT. As described by the authors, a longer time between LT and KT by itself might not be a risk factor, but a result of a complicated situation.   Furthermore, kDGF might not be a risk factor for postoperative complications, but actually be a ‘marker’ of hepatic ischemia/reperfusion injury, resulting in a complicated postoperative course.”

In response, in order to emphasize these points, we did 2 things.  First, in the 2nd paragraph of the Discussion, where we state that a longer time between LT and KT by itself might not be a risk factor, we added a clarifying sentence (using some of your wording – thanks!) at the end of that paragraph as follows: “Similarly, kDGF might not by itself be a risk factor for the development of postoperative complications but actually could be a marker of hepatic ischemia/reperfusion injury, resulting in a more complicated postoperative course.”  Second, in the study limitations paragraph (towards the end of the Discussion) we now state (again, using some of your wording – thanks!), “There are several limitations to this study, particularly that of cause and effect.  As previously described, patients with a difficult LT procedure might first have to be stabilized before they can receive the KT; thus, a longer time between LT and KT by itself might not be a risk factor, but a result of a complicated situation.” 

2. “Can the authors describe the protocol regarding timing of KT in combined LKT. Is there a standard period?  And what are the reasons for delaying a KT?”

In response, we added the following clarifying paragraph at the end of the Discussion as follows: “Currently, at the Miami Transplant Institute, we do not have a standard protocol in place that determines the optimal timing of KT after LT during CLKT.  By performing this retrospective study, our hope was to identify potential variables (such as the requirement for PBRCs during LT and post-LT requirement of vasopressor support) that could subsequently be used in establishing a more uniform approach to this growing group of solid organ transplants recipients.”

3. “The Introduction is relatively long, i.e., a detailed description of HPMP can be given in the Methods or Discussion.”

In response we shortened the Introduction by removing 4 long sentences that we thought were unnecessary to retain in the Introduction.  We also rewrote and reordered a number of other sentences; we believe that the Introduction reads much better now.”

4. “There are multiple writing errors in the text, correction for English language is required.”

In response we have gone through the whole manuscript and rewrote numerous sentences in which the English/grammar was not so clear.”

5. “’Primary outcome’ is used for the comparison of interventions in clinical trials, not observational studies. The conclusion kDGF is ‘responsible’ for … is somewhat premature for the results presented in this retrospective study.”

With all due respect, the first part of this comment is simply untrue, as all clinical studies, whether they are retrospective in nature, single arm prospective, or multiple arm randomized trials, contain a primary purpose along with secondary goals.  Usually, there is a primary endpoint that is analyzed to address the primary study purpose; secondary endpoints are analyzed to address the secondary purposes of the study. However, in an attempt to address the reviewer’s concern on this matter, we now use the terms, “primary study endpoint” and “secondary study endpoints in the Materials and Methods section.  Regarding the second part, we agree with the reviewer’s concern, and in response we changed in the Abstract “is responsible” to “may be responsible.”

6. “Please delete the table with the included variables in this study. This could be described in the text.  The variables will be shown in the tables with the results already.”

In response we agree that this table was unnecessary to include, so it was removed.”

7. “Regarding machine perfusion: All kidney were perfused with HPMP. Was this oxygenated perfusion, and what kind of solution was used, please describe KPS.  Were the livers in these patients also perfused?  Please describe.  As perfusion of the liver might decrease the hepatic ischemia/reperfusion injury, this might also affect injury to the kidneys in the early post-transplant period.”

In response we added the following clarifying paragraph in the Materials and Methods section (paragraph 2.2): ”Following kidney recovery, all deceased donor kidneys were connected to the LifePort Renal Preservation Machine® (Organ Recovery Systems, Itasca, IL, USA), a type of non-oxygenated hypothermic machine perfusion. Renal allografts were arterially cannulated and perfused with Kidney Perfusion Solution® (KPS-1). KPS-1, having the same composition as UW® Machine Perfusion Solution, originally formulated at the University of Wisconsin, is a clear, sterile, non-pyrogenic, non-toxic solution with an osmolality of 300 ± 15 mOsm/kg, a sodium concentration of 100 mEq/L, a potassium concentration of 25 mEq/L, and a pH of 7.4 ± 0.1 at room temperature. Based on the sodium/potassium ratio, the composition is thus consistent with that of an extracellular solution.”

And, at the end of the same paragraph, we added: “Hypothermic perfusion was not applied to liver allografts.”

8. “Can the authors also describe the technicalities of the liver transplant procedure? Classic, piggyback, veno-venous bypass?  This could all affect kidney function after LT.”

In response we added the following clarifying paragraph in the Materials and Methods section (paragraph 2.3): ”All LT cases in our series were performed using the piggyback hepatectomy technique without employing a veno-venous bypass.”

9. “The authors describe a standard second look in their liver transplant patients, is this also for liver only patients? If any intervention is performed during this procedure, does it count as a postoperative complication?”

In response à The second look operation after LT is a standard protocol at our Center, both during isolated LT and CLKT, since at the time of liver transplantation only the skin is closed as described in the Materials and Methods Section. When a standard second look operation is performed consisting only of an accurate washout of the abdomen and fascia closure, this was never considered as a postoperative complication. However, if during this second look operation, any type of intervention different from those above mentioned was performed, for example a re-do of the biliary anastomosis, it was considered as a postoperative complication classified as a grade IIIb according to Clavien-Dindo scale. A second look operation for the kidney side is never performed unless there is a clear indication to perform this procedure (hemorrhage, collections, thrombosis etc…). Therefore, we added the following clarifying paragraph in the Materials and Methods section (paragraph 2.3): “By performing the standard second-look operation after LT as described above, it was not automatically included among the postoperative complications, as this approach represents a surgical protocol at our Institute. However, if any other procedures in addition to those listed above were performed, then the second look operation was classified as a grade IIIb postoperative complication, according to the Clavien-Dindo scale.” And, subsequently in the same section, we added: “Primary fascia closure was always accomplished at the end of KT.”

10. “The authors describe the immunosuppression starting at KT. Did the patients not receive their first immunosuppression when the liver was transplanted”

In response we added the following clarifying paragraph in the Materials and Methods section at the end of paragraph 2.3: ”The first induction dose of methylprednisolone alone was given intraoperatively before reperfusion of the liver allograft, whereas the first induction doses of antithymocyte globulin and basiliximab were administered intraoperatively before reperfusion of the renal allograft.”

11. “The table with 5 multivariable analyses is very confusing and has limited value. The authors just summarized the results but failed to describe the additional value of each analysis.”

In response à we added some clarifying sentences into the last paragraph of the Multivariable Analysis Results subsection as follows: “Lastly, kDGF was a highly significant multivariable predictor of a higher CCI, even after controlling for the 4 other significant multivariable predictors of this outcome (P<0.000001, Table 5c).  While kDGF was not an important multivariable predictor of the development of hospital-acquired infections (P=.05, Table 5d), kDGF was a highly significant multivariable predictor of a poorer renal function at 1-month post-transplant (i.e., eGFR<45 ml/min/1.73m2), even after controlling for the 2 other significant multivariable predictors of this outcome (P<0.000001, Table 5e).  Thus, the important univariable and multivariable associations of kDGF with less favorable clinical outcomes (as shown in Tables 4 and 5, respectively) were similar.”  In addition, we now designate Tables 5a-5e to distinguish among the 5 multivariable models that were presented in Table 5 (each one presented with the main purpose of determining the prognostic impact of kDGF in each model).

Reviewer 2 Report

Vincenzi et al. provide recent data regarding combined liver-kidney transplantation in a large single-center cohort of 115 patients.

The article is very informative and well-written. 

I have only minor issues:

-recommended terminology from the International Club of Ascites (Angeli et al., Gut, 2015) is not used

-why were patients with HRS given CLKT instead of standard OLT?

-BMI was used, although this is a variable heavily biased in patients with ESLD (and CKD). It would be preferable to use a measure of sarcopenia before drawing conclusions between kDGF and low BMI (page 14 line 445).

-no software is mentioned in the Methods section.

Author Response

Point-by-Point Responses to the Reviewers’ Comments

Reviewer #2’s Comments

1. “Recommended terminology from the International Club of Ascites (Angeli et al, Gut 2015) is not used.”

In response, thanks for referring to this precious paper on the diagnosis and management of acute kidney injury in cirrhotic patients. Indeed, at our center, the transplant nephrologists follow the International Club of Ascites (ICA) guidelines for the diagnosis and staging of AKI in cirrhotic patients independently from the etiology. According to this, we added the following clarifying sentence in the Materials and Methods section at the end of paragraph 2.1: “International Club of Ascites (ICA) criteria were used for the diagnosis and staging of acute kidney injury (AKI), regardless of the cause.”

2. “Why were patients with HRS given CLKT instead of standard OLT?”

In response, as reported in the Materials and Methods Section, precisely paragraph 2.1, we decided to include patients transplanted since 2015 based on the adoption of uniform and homogenous criteria for CLKT listing at our Institute, also in agreement with those introduced by UNOS in 2017. These criteria allow listing for CLKT in patients with sustained acute kidney injury, even in the setting of cirrhotic patients where the AKI is generally secondary to HRS, when specific conditions are met such as dialysis for > 6 weeks, GFR £ 25 ml/min for > 6 weeks documented every 7 days or combination of these two criteria for at least 6 weeks of duration.

According to this, we added the following clarifying sentence at the beginning of Materials and Methods section: “These criteria apply to both cirrhotic patients with chronic kidney disease (CKD) and with acute kidney injury (AKI), that in the setting of chronic liver failure is mainly sustained by type 1 hepatorenal syndrome (HRS).”

3. “BMI was used, although this is a variable heavily biased in patients with ESLD (and CKD). It would be preferable to use a measure of sarcopenia before drawing conclusions between kDGF and low BMI (page 14, line 445).”

In response, we added the following clarifying sentence at the end of the paragraph in question in the Discussion section, “Clearly, it would be preferable to have included a measure of sarcopenia, before drawing any firm conclusions between kDGF and low BMI.”

4. “No software is mentioned in the Methods section.”

In response, we added the following sentence as the last sentence in the Statistical Analysis subsection in the Materials and Methods section as follows: “All of the statistical analysis was performed using the Statistical Analysis System (SAS) software, Version 9.4, SAS Institute, Inc., Cary, NC.”